# Factors Influencing Functional Recovery during Rehabilitation after Severe Acquired Brain Injuries: A Retrospective Analysis

Paolo Finotti [1], Massimo Iannilli [1], Lucrezia Tognolo [2],*, Claudia Vargiu [2], Stefano Masiero [2]
and Giovanni Antonio Checchia [1]

[1] Recovery and Functional Rehabilitation Unit, Hospital of Conselve, 35026 Padua, Italy;
finottipaolo89@libero.it (P.F.); massimo.iannilli@aulss6.veneto.it (M.I.);
giovanniantonio.checchia@aulss6.veneto.it (G.A.C.)

[2] Department of Neurosciences, Physical Medicine and Rehabilitation School, University of Padua,
35128 Padua, Italy; claudiavargiu92@gmail.com (C.V.); stef.masiero@unipd.it (S.M.)

* Correspondence: lucrezia.tognolo@unipd.it; Tel.: +39-049-8213-353

**Abstract:** Severe acquired brain injuries (sABI) represent one of the main causes of disability and limitation in social life participation that need an intensive rehabilitation approach. The purpose of this study was to identify a possible correlation between different supposed conditioning factors and the efficiency of rehabilitation interventions. In this retrospective study, data were processed regarding 44 patients admitted to a neurorehabilitation department after sABI. A significant correlation with the efficiency of the rehabilitation intervention (expressed as the variation of the Barthel score between discharge and admittance in relation to the duration of the rehabilitative hospitalization) was found for both the etiology of the brain injury ($p = 0.023$), the precocity of the rehabilitation treatment ($p = 0.0475$), the presence of a tracheal cannula ($p = 0.0084$) and forms of nutrition other than oral ($p < 0.0001$). The results of this study suggest that improving the management of the respiratory system, swallowing and nutritional aspects, and favoring an early and personalized rehabilitation treatment, can help to optimize the overall care of patients suffering from sABI, thus allowing a reduction in complications, improvement in functional recovery and ensuring a better management of economic, social and health resources.

**Keywords:** severe acquired brain injury; rehabilitation; functional recovery; conditioning factors; nutrition; tracheostomy; infections

## 1. Introduction

A severe acquired brain injury (sABI) consists of brain damage due to cranioencephalic trauma or other causes that determines a protracted condition of coma (lasting more than 24 h) and sensorimotor, cognitive or behavioral impairments, leading to disability.

Due to the different spectrum of residual impairments and disabilities, and to the different incidence rates in the various age groups, it is important to distinguish between traumatic and non-traumatic brain injuries. The latter can originate from brain tumors, cerebral anoxia, severe hemorrhagic syndromes, infections (encephalitis) and toxic-metabolic encephalopathies [1,2].

Those affected require appropriate interventions in the emergency phase, a prompt hospitalization in a suitable location for intensive care and neurosurgical treatments, lasting from a few days to a few weeks (acute phase) [3,4].

Afterwards, intensive medical-rehabilitation interventions are usually necessary and need to be carried out as soon as possible in hospitalization, which can last from a few weeks to a few months (post-acute phase) [5].

In most cases, after the hospitalization, long-term health and social interventions are necessary and are aimed at addressing the outcomes. [6,7]. For the appropriate management of a patient with sABI, the intervention of a multi-disciplinary and multi-professional

team is crucial. The rehabilitation team must carefully assess neurological damage and sensorimotor, cognitive and behavioral impairments, which together significantly affect the functional abilities of the patient, resulting in disabilities of varying severity [8].

Metabolic and nutritional alterations also need particular attention as a high percentage of subjects (more than a third of those admitted to rehabilitation centers) require artificial feeding, which is generally administered through percutaneous endoscopic gastrostomy (PEG) [9,10].

The management of the tracheostomy and the decannulation process represents another priority problem to be addressed in high intensity rehabilitation units. Tracheostomy is often indicated in the acute phase of sABI as it favors mechanical ventilation, reduces respiratory resistance, decreases the need for sedation, reduces intubation times, controls the risk of inhalation, promotes the management of bronchial secretions and ensures the patency of the airways. Conversely, the presence of a tracheal cannula induces significant discomfort for the patient, makes verbal communication impossible or difficult, increases the risk of infections, reduces the normal larynx movements and complicates the dynamics of swallowing (usually already compromised by the brain injury itself) [11,12].

The purpose of this retrospective study was to evaluate whether the etiology of sABI, the time elapsed between the acute event and the rehabilitation hospitalization, the presence of a tracheal cannula, the feeding modalities and the number of infectious episodes could represent possible conditioning factors able to influence the efficiency of the rehabilitation intervention. The hypothesis was that several factors can determine not only a worse functional recovery but also a prolonged hospitalization with a more expensive care management, causing inefficient management of the socio-economic and assistance resources available. Moreover, as a secondary endpoint, the study aimed to the analyze possible correlations between the etiology of sABI and patients' sex and age, the number of infectious episodes, the total duration of the rehabilitative hospitalization, the presence of tracheostomy and the feeding modality.

## 2. Materials and Methods

In this observational retrospective study, data relating to all the patients admitted to the neurorehabilitation department of the Conselve Hospital (Padua, Italy) from 1 January 2015 and discharged by 30 June 2020, and suffering from sequelae of severe acquired brain injury, were collected and processed. Inclusion criteria: age > 18 years, Glasgow Coma Scale after the initial insult < 9, and arrival in the neurorehabilitation department less than one year after the acute event. Exclusion criteria: additional spinal cord injury and the need for ventilation support.

Specifically, the following data was collected for each subject: age, sex, etiology of the brain injury (traumatic or non-traumatic), time between the acute event and the rehabilitative hospitalization, presence of a tracheostomy at admittance, type of nutrition at admittance and discharge, number of infectious episodes that occurred during the hospitalization, score on the Barthel Index at admittance and discharge, and the total duration of the rehabilitative hospitalization.

This data was processed to assess a possible correlation between different supposed conditioning factors and the efficiency of the rehabilitation intervention. For this purpose, the ratio between the functional gain (expressed as the variation of the score on the Barthel Index between discharge and admission) and the duration of the rehabilitative hospitalization was calculated and used as an index of efficiency of the rehabilitation intervention.

The etiology of sABI, the time elapsed between the acute event and the rehabilitation hospitalization, the presence of a tracheal cannula, the feeding modalities and the number of infectious episodes were investigated as potential conditioning factors. For each of these factors, the correlation with the aforementioned index of efficiency of the rehabilitation intervention was analyzed (primary endpoint).

In addition, further correlations were analyzed as secondary endpoints. In particular, the etiology of sABI was also correlated to sex and age, while the number of infectious episodes was correlated to the total duration of the rehabilitative hospitalization. Finally, the correlation with the incidence of infectious episodes was analyzed for both tracheostomy and feeding modality.

The Barthel Index is a scale with excellent validity, reliability and sensitivity. It is very widespread in rehabilitation departments and in the scientific literature. Its main purpose is to establish the degree of independence and to assess a person's disability and care needs. It is an ordinal scale with a total score from 0 (totally dependent subject) to 100 (totally independent), divided into ten items (each with specific scores): nutrition, bathing, personal hygiene, dressing, rectum and bladder control, transfer to the bathroom and chair/bed, walking and climbing stairs [13].

*Statistical Analysis*

Quantitative data were summarized with mean and standard deviation, median, minimum and maximum, while for qualitative data the number and percentage of subjects in each category were reported.

Various groups were identified, and inter- and intra-group analyses were carried out.

The comparison of the quantitative variables between groups was performed with the Wilcoxon test for independent samples, while the comparison of the qualitative variables between groups was conducted with Fisher's exact test.

The correlation between quantitative variables was evaluated with the Spearman correlation coefficient and its statistical significance.

The level of statistical significance was set at 0.05 for all analyses.

## 3. Results

All of the patients admitted in the neurorehabilitation unit of Conselve between January 2015 and June 2020 were enrolled in the retrospective analysis. The study included 44 subjects (29 male, 15 female), aged between 37 and 78 years (with a mean age of 59.5 years).

The average duration of the rehabilitative hospitalization was 101.45 days ($\pm$66.5 days), with a minimum of 8 days and a maximum of 354 days. A subject died after 24 days from the admission to the neurorehabilitation unit due to a cardio-circulatory arrest. As the Barthel Index was repeated weekly, that patient was also included in the analysis using the last Barthel score before death as the Barthel at discharge.

A traumatic etiology was the cause of sABI in 14 subjects, while among the remaining 30 cases with non-traumatic etiology, the main cause was a hemorrhagic event (21 cases), followed by ischemic stroke (8 cases) and hypoxic encephalopathy after cardiac arrest (1 case). Table 1 shows the data relating to age, sex and the efficiency of the rehabilitation intervention, divided into two groups based on the etiology of sABI (traumatic or non-traumatic).

The duration of hospitalization in the acute care units (i.e., the time elapsed between the acute event and the admission to neurorehabilitation unit) was found to be an average of 61.86 days (with a standard deviation of 35.70 days and a median of 57 days). A negative ($-0.304$) and statistically significant ($p = 0.0475$) correlation emerged between this variable and the efficiency of the rehabilitative hospitalization (Figure 1).

The initial severity of the brain damage was similar among the patients, as all of them had a Glasgow Coma Scale between 3 and 6 after the initial insult. Similarly, the Barthel score on admission to the neurorehabilitation unit was 0 or 5 for all subjects.

**Table 1.** Data analysis in relation to the etiology of sABI is distinguished into non-traumatic and traumatic. The first column shows the analyzed variables: age, sex and efficiency of the rehabilitation intervention (expressed as the variation of the Barthel score between discharge and admittance in relation to the duration of the rehabilitative hospitalization). The last column describes the *p*-value of the comparison between groups. SD = standard deviation, MIN = minimum, MAX = maximum, F = female, M = male.

| Variable | Non Traumatic (N = 30) | Traumatic (N = 14) | *p*-Value |
|---|---|---|---|
| Age Mean ± SD Median (MIN-MAX) | 61.77 ± 9.81 62.50 (39.00–78.00) | 54.64 ± 13.73 56.00 (37.00–76.00) | 0.11 |
| Sex F M | 13 (86.7%) 17 (58.6%) | 2 (13.3%) 12 (41.4%) | 0.089 |
| Delta Barthel/days in neurorehab Mean ± SD Median (MIN-MAX) | 0.42 ± 0.66 0.14 (0.00–2.38) | 0.81 ± 0.71 0.55 (0.00–2.21) | 0.023 |

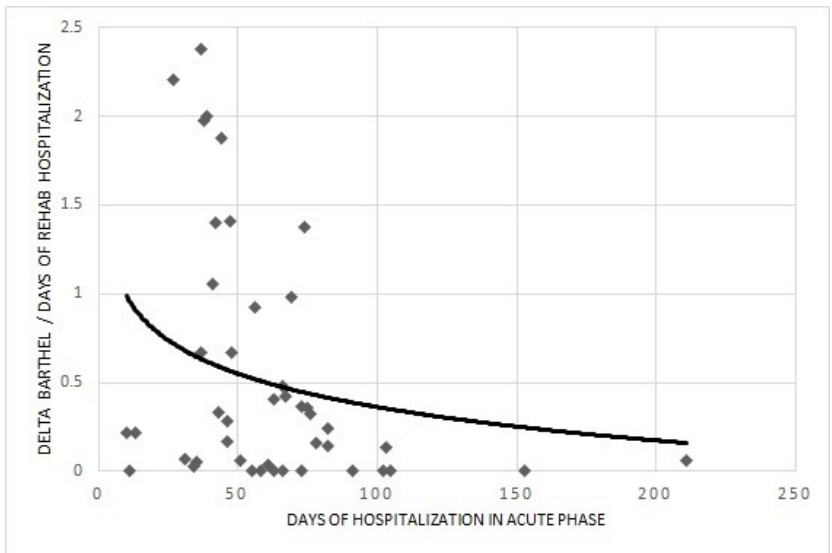

**Figure 1.** Correlation analysis between the time elapsed from the acute event to the rehabilitation hospitalization (x-axis) and the following efficiency of the rehabilitation intervention (y-axis).

In our sample, 26 subjects were admitted to the neurorehabilitation unit with a tracheostomy. None of them were still dependent on respiratory ventilation. Table 2 summarizes the data analysis carried out by distinguishing the subjects into two groups based on the presence or absence of the tracheostomy at the time of admission to the neurorehabilitation department.

Only 36.8% of patients with tracheostomy at the time of admission did not experience infectious episodes that required antibiotic treatment during the rehabilitation hospitalization, compared to 63.2% of patients without cannulas. Twelve subjects were admitted to neurorehabilitation with an indication to follow isolation procedures for infections already identified in the acute care unit, while a total of another 36 infectious episodes developed during the rehabilitation hospitalization. Of these 36 infections, 14 were respiratory ones; only one of them occurred in a patient without tracheostomy (respiratory infection from SARS-CoV-2) and the other 13 in tracheostomy carriers. The other infections were urinary, blood or gastrointestinal ones (or combined infections).

**Table 2.** Analysis of the data relating to the presence or absence of the tracheostomy at the time of admission to neurorehabilitation. The first column shows the analyzed variables; for infections, data are reported as the mean (with standard deviation) and median number of infections of any type per subject. The second and third columns report the data relating to patients admitted to the neurorehabilitation unit without or with tracheostomy, respectively, while the last column describes the *p*-value of the comparison between groups. tracheo = tracheostomy, SD = standard deviation, MIN = minimum, MAX = maximum.

| Variable | Tracheo = No (N = 18) | Tracheo = Yes (N = 26) | *p*-Value |
|---|---|---|---|
| Infections | | | |
| Mean ± SD | 0.20 ± 0.41 (N = 15) | 1.00 ± 1.03 (N = 18) | 0.011 |
| Median (MIN-MAX) | 0.00 (0.00–1.00) | 1.00 (0.00–6.00) | |
| Delta Barthel/days in neurorehab | | | |
| Mean ± SD | 0.89 ± 0.81 | 0.30 ± 0.48 | 0.0084 |
| Median (MIN-MAX) | 0.67 (0.00–2.38) | 0.14 (0.00–1.98) | |

The data presented in Table 3 were obtained by evaluating the number of infectious episodes that occurred during the hospitalization in the neurorehabilitation unit, and distinguishing the subjects who presented two or more infectious episodes from those who presented one or none.

**Table 3.** Data analysis in relation to the number of infectious episodes that occurred during the hospitalization in the neurorehabilitation unit. The first column describes the analyzed variables (length and efficiency of the rehabilitation hospitalization), while the following columns show data for subjects who presented one or zero infectious episodes, two or more infections, and the *p*-value of the comparison between groups, respectively. SD = standard deviation, MIN = minimum, MAX = maximum.

| Variable | Infections = 0−1 (N = 34) | Infections = 2+ (N = 10) | *p*-Value |
|---|---|---|---|
| Days of rehab hospitalization | | | |
| Mean ± SD | 86.26 ± 47.71 | 153.10 ± 94.47 | 0.030 |
| Median (MIN-MAX) | 91.00 (8.00–168.00) | 110.00 (43.00–354.00) | |
| Delta Barthel/days in neurorehab | | | |
| Mean ± SD | 0.60 ± 0.71 | 0.37 ± 0.64 | 0.086 |
| Median (MIN-MAX) | 0.32 (0.00–2.38) | 0.01 (0.00–1.98) | |

Only 11 subjects reached the neurorehabilitation unit already with complete oral feeding; among the remaining 33 patients, 24 were fed with PEG.

Table 4 reports the processing of data relating to the presence or absence of a complete oral feeding at the time of admission to the rehabilitation department.

Of the 33 subjects who reached the neurorehabilitation unit with alternative forms of nutrition, 18 were discharged with complete oral feeding, 14 with the need for feeding or supplementation by other means (in the vast majority of cases, through PEG) and one patient died.

Statistically significant differences ($p = 0.0002$) in terms of efficiency of the rehabilitation intervention (Delta Barthel/days in neurorehabilitation) were found among the patients who were weaned from alternative feeding methods during the rehabilitative hospitalization and those who were discharged without complete oral feeding.

**Table 4.** Data analysis based on the type of nutrition at the admittance to the rehabilitation unit. Among the analyzed variables (first column), data concerning infections are reported as the mean (with standard deviation) and median number of infections of any type per subject. The second and third columns show the data relating to patients without or with complete oral feeding at the time of admission, respectively, while the *p*-value of the comparison between groups is reported in the last column. SD = standard deviation, MIN = minimum, MAX = maximum.

| Variable | Oral Nutrition = No (N = 33) | Oral Nutrition = Yes (N = 11) | *p*-Value |
|:---:|:---:|:---:|:---:|
| Infections Mean ± SD Median (MIN-MAX) | 0.83 ± 0.96 (N = 24) 1.00 (0.00–6.00) | 0.11 ± 0.33 (N = 9) 0.00 (0.00–1.00) | 0.027 |
| Delta Barthel/days in neurorehab Mean ± SD Median (MIN-MAX) | 0.29 ± 0.45 0.14 (0.00–1.98) | 1.29 ± 0.76 1.38 (0.22–2.38) | <0.0001 |

## 4. Discussion

In our study, the efficiency of rehabilitation intervention was higher for patients with traumatic etiology than for those with non-traumatic etiology. This means that, in relation to a comparable expenditure, the functional recovery was greater in the non-traumatic group.

Similarly, various other studies reported better outcomes at discharge for patients with traumatic ABI compared to those with non-traumatic etiology [14,15].

The better outcome levels reported by traumatic patients may be related to the nature of the initial impairment: the damage is often diffuse in non-traumatic forms, while in traumatic forms, large parts of the brain are often undamaged, providing a neuronal reserve for adaptive neuroplastic changes [16].

A further possible explanation may be related to the impact of age; prior investigations have shown that, despite similar early improvements in functioning, younger patients are more likely to show continued improvement over time, while older patients are more likely to decline. Usually being older, non-traumatic patients more often present pre-morbid pathologies, social and financial disadvantages and a poor family network, causing a significant impact on functional recovery and on home assistance for care needs [17].

In agreement with previous literature, our analysis shows a lower (albeit not statistically significant) average age of about 7 years (61.77 versus 54.64) in the traumatic group compared to the non-traumatic group. Considering that the most common aetiologies of traumatic brain injuries are represented by falls, vehicular accidents, sports and violence, while those for non-traumatic forms consist mainly of stroke and anoxia following cardiac arrest, may explain the younger age in the traumatic group [18].

Moreover, our data highlight a higher incidence of sABI in males, with an overall percentage of 65.9% of males, compared to 34.1% of females. The prevalence of males appears even more pronounced (albeit not statistically significant) in patients with traumatic sABI, reaching as much as 85.7%. These data are almost in line with what was reported by Chiavaroli and collaborators, who documented a greater probability of experiencing sABI in males than females, especially for traumatic forms. [19] According to some authors, the higher incidence of traumatic brain injuries among men could be associated with higher risk-taking activities, including occupational ones, and higher rates of violence than women [20].

A statistically significant negative correlation emerged from the comparison between the time elapsed from the acute event to the admission to the neurorehabilitation unit and the following efficiency of the rehabilitative hospitalization. Indeed, subjects who arrived later to the rehabilitation department showed minor scores on the index of rehabilitative intervention efficiency. This aspect can be explained by the significantly better outcomes associated with early rehabilitation, as reported by many other studies [21].

A long hospitalization in intensive care or other acute care units is, in fact, correlated with various complications and linked to long bed rest periods that involve various negative effects on the nervous, musculoskeletal, pulmonary, cardiovascular and metabolic systems [22].

Several studies underline the importance of physiotherapy treatments in intensive care units [23–25]. Indeed, physical activity stimulates the release of neurotrophic substances, which promote the metabolism of neuronal cells and the growth of nerve fibers [26]. Early rehabilitation can therefore affect brain plasticity and consequently accelerate the recovery process [27–29]. Consequently, early rehabilitation is associated with an improved outcome in terms of reduction in the duration of comas and hospitalization, better cognitive level at discharge, better score on the functional scales and greater chances of returning home [30,31].

There is also a positive correlation between the intensity of the rehabilitation treatment and the outcome, which once again highlights how early access to an intensive rehabilitation department can allow a greater functional recovery [32,33].

In our sample, the average duration of the hospitalization in the acute care units is surely longer than that reported in literature. The delay in the access to the neurorehabilitation unit of Conselve Hospital is due to its location (several kilometers from the central hospital) and to the absence of an intensive care unit for emergency support. For this reason, many patients with persistent autonomic instability stay in the Hub Hospital for a prolonged period, with a delay in access to the rehabilitation unit compared to other centers located in the Hub Hospitals. Various authors have observed a relevant impact of the clinical complexity of patients with sABI on early admission to the rehabilitation setting, due to medical or surgical complications and the frequent need to transfer them from rehabilitation units to acute care ones. In their opinion, the poor outcome of these subjects is more likely to be related to more severe brain damage rather than a late rehabilitation hospitalization. [34]

The processing of our data reported significantly better scores in terms of the efficiency of the rehabilitation intervention for subjects who arrived in the neurorehabilitation ward without tracheostomy (mean score of delta Barthel/days of rehabilitation hospitalization of 0.89 compared to 0.30 of tracheostomy carriers).

Similarly, other studies have provided some evidence about the influence of the respiratory status as a relevant prognostic factor in those who have suffered from sABI, reporting higher rates of home discharge and better functional outcomes in the absence of tracheostomy [35,36].

In addition, a significantly higher incidence of infectious episodes with various origins (particularly respiratory ones) was found in our study among subjects with tracheostomy.

Several works underline how infections, especially those involving the respiratory tract, represent a common complication in subjects suffering from sABI, as well as a factor associated with poor functional results even several years after the injury [37].

Furthermore, in our study sample, a greater number of infectious episodes (two or more compared to one or none) was associated with a significantly longer rehabilitation hospitalization (153.10 vs. 86.26 days) and a worse index of efficiency of rehabilitative intervention (although not statistically significant).

Patients with brain injuries are often unable to feed orally due to cognitive impairments, dysphagia, assisted ventilation or other conditions [38].

As for the presence of a tracheostomy tube, in our study sample, the need for a PEG, nasogastric tube or intravenous feeding was also associated with a greater risk of infections and a worse efficiency of rehabilitative intervention.

Although associated with worse results compared to the subjects already weaned from alternative nutrition in the acute care unit, the resumption of complete oral feeding in the neurorehabilitation unit was associated with a better index of rehabilitation efficiency than in patients forced to maintain alternative forms of feeding even upon discharge.

The return to oral feeding therefore represents an important rehabilitation goal, albeit not always achievable, because there is a significant positive correlation between the start of oral feeding and the functional outcome in patients with sABI [39].

The return to oral feeding is often linked to the possibility of decannulation from tracheostomy, which must take place following standardized guidelines [29,30,40]. A failure in the decannulation process is usually associated with a constant need for alternative feeding, as well as an extremely reduced functional recovery with important social-health costs and assistance needs even after the discharge from the intensive rehabilitation department.

This work undoubtedly has several limitations. First, the retrospective study model, the small sample size and the presence of numerous confounding factors increases the risk of possible biases. Furthermore, we have evaluated the efficiency of the rehabilitation intervention through an index that does not have accurate demonstrations of scientific validity in literature. It would have been useful to collect other validated scales of functional assessment in order to compare the results obtained and confirm our findings.

In spite of such limitations, this study can provide useful indications about the epidemiology, the conditioning factors and the rehabilitation efficiency relating to subjects with severe acquired brain injury.

The management of these patients in a specialized rehabilitation unit represents an extremely demanding process, which requires important resources as well as a complex and experienced care and rehabilitative organization. In particular, the rehabilitation team must guarantee an accurate prognostic classification, establish personalized rehabilitation objectives and select the most suitable therapeutic-assistance paths [30].

While some prognostic factors (for example, age, sex and etiology of brain injury) are not modifiable, it is possible to actively intervene on other factors in order to improve the recovery process and the functional outcome. A fundamental element appears to be the correct management of the respiratory condition to guarantee a safe and timely decannulation from the tracheostomy; similarly, an accurate management of the nutritional situation is crucial in order to reduce the infectious risk, avoid the risk of malnutrition and allow the subject to return to oral feeding as early as possible.

The originality of this study is that, while the prognostic factors of sABI are well known, their impact in terms of efficiency of the rehabilitation intervention has not been investigated in literature. Our results suggest that, by carefully intervening on the aforementioned aspects, it is possible to achieve not only a greater functional recovery but also a better management of the economic and social-health resources through the acceleration of the recovery process.

## 5. Conclusions

Severe acquired brain injuries represent one of the main causes of physical, cognitive and psychological disability and limitation to social participation. The complexity of assessment, therapeutic, rehabilitative and care needs require specialized and experienced organization.

A multi-disciplinary and multi-professional team must take care of the patient as early as possible, define accurate prognoses and fully understand the factors that influence the rehabilitation trajectory, in order to be able to realize a personalized and efficient rehabilitation program.

Adequate management of respiration and tracheostomy, swallowing and feeding modalities, associated with an early and intensive rehabilitation treatment, allows for a reduction in complications, improving the functional recovery with a shorter hospitalization and consequently guaranteeing a better management of economic, health and social resources.

**Author Contributions:** P.F.: investigation; writing—original draft preparation; M.I.: conceptualization, methodology; L.T.: writing—review and editing; C.V.: investigation; S.M.: supervision; G.A.C.: conceptualization, supervision. All authors have read and agreed to the published version of the manuscript.

**Funding:** This research received no external funding.

**Institutional Review Board Statement:** Ethical review and approval were waived for this study, due to the observational retrospective feature of the study.

**Informed Consent Statement:** Informed consent was obtained from all subjects involved in the study.

**Conflicts of Interest:** The authors declare no conflict of interest.

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
