# Peer review of "Factors Influencing Functional Recovery during Rehabilitation after Severe Acquired Brain Injuries: A Retrospective Analysis"

_traumacare, doi:10.3390/traumacare1030015_

Round 1

Reviewer 1 Report

Aim of the presented study was to evaluate factors that may influence the functional recovery of patients with severe acute brain injury and to evaluate the efficiency of the therapeutic intervention during the rehabilitation. 44 patients with traumatic and non-traumatic injury were evaluated. The influence of hospitalization in the acute care units, the presence of tracheostomy, the need for non-oral feeding and the number of infections were the parameters that were statistically analysed. From their retrospective data analysis the authors conclude that optimizing respiratory, swallowing and nutritional aspects can help to improve functional recovery.

Comments

How many patients were admitted to the Neurorehabilitation department between 2015 and 2020 in total? How many of these patients had severe brain injury? Were all patients with severe brain injury included in the study? How many had to be excluded? No inclusion or exclusion criteria are presented.

Statistical analysis: Cannot be really understood. The Kruskal-Wallis test was used for comparison between three groups. However, in the results there are only comparisons of two groups. The whole sentence cannot be really understood.

The average duration of the rehabilitation was 101 days. Please add standard deviation.

Was the patient, who died, omitted? If not, how can he be included e.g. in the delta Barthel? Are the data presented on 43 or 44 patients?

It makes no sense to include a patient with hypoxic brain damage into the study (spelling error page 4: hypossic), as this patient has no potential for rehabilitation.

The presentation of the data is insufficient. First (table 1) traumatic and non-traumatic patients are regarded separately. Then infections, nutrition and tracheostomy are analysed regardless of the underlying pathology. Trauma patients and vascular patients have to be regarded completely different as they do not share the same prognostic baseline. Trauma patients may have additional injuries that can negatively influence recovery (e.g. head trauma with additional spinal cord injury).

No data about the initial severity of the brain damage is given. Please add GCS after the initial insult to the patients’ characteristics. In addition to this information table 1 should also include at least the number of days in the ICU, the Barthel on admission and discharge from the rehab-centre, the number of patients with tracheostomy on admission and the number of patients with oral feeding.

Figure 2: The average day on the ICU with 62 days seems extremely long. This is unusual and seems unrelated to the disease. What is the real reason? There seems to be a lack of rehabilitation centres, which could take the patients at an earlier point of time. Needs to be discussed. Please give the median of ICU treatment and the range. The correlation coefficient of -0.3 dose not really show an association (should be >-0.5, the p-value does not say anything about the correlation, it just tells us that the calculated value is significant) and might be influenced by the two patients with more than 150 and 200 days of ICU treatment. What happens, when these statistical outliers are eliminated?

Table 2: What type of infections? Respiratory (which the reader assumes with the tracheostomy), urinary, blood? All type of infections? Please specify. How many of these patients were still dependent on respiratory ventilation?

The number of infections is different in table 2 and 4. The data set is always 44 patients and it is assumed, that the same data set for infections is used. In table two the maximum number of infections in any patient is six. In table 4 the maximum number of infections in any patient is three. Explanation?

The calculation of the mean number of infections in table 2 cannot be followed. How was it calculated? The reader assumes that the numbers of infections per patient (0, 1, 2, 3 etc.) were summed up, and that this whole number was then divided by the number of patients (n=18). Why does multiplication of 0.2x18 not result in a whole number but 3.6? The same problem can be found in table 4 (e.g., 0,11x11=1,21, which is not a whole number).

Table 3: What type of infections were observed besides respiratory infections?

There is a lot of confounding parameters which is statistically not analysed. The data show a better effect of treatment in patients without tracheostomy. Is this really surprising? Patients with tracheostomy have more infections. To be expected and well known. That these patients will stay longer in the hospital can also be expected.

Discussion: The most common cause for traumatic brain injury is fall.

The conclusions presented is common knowledge and, as presented, not supported by the data. A correlation of efficiency of the rehabilitation is not presented (as indicated in the abstract). The data show, that the patients have a different daily improvement when different aetiologies, patients with or without infections and patients with different forms of nutrition are compared.

Author Response

We thank you the Reviewer for the observations. Below, we explain point-by-point the details of the revisions and the responses to the comments.

How many patients were admitted to the Neurorehabilitation department between 2015 and 2020 in total? How many of these patients had severe brain injury? Were all patients with severe brain injury included in the study? How many had to be excluded? No inclusion or exclusion criteria are presented.

All the patients admitted to the Neurorehabilitation Unit of Conselve between 2015 and 2020 were enrolled in the retrospective analysis. All of them had severe brain injuries and no one was excluded. We added inclusion and exclusion criteria.

Statistical analysis: Cannot be really understood. The Kruskal-Wallis test was used for comparison between three groups. However, in the results there are only comparisons of two groups. The whole sentence cannot be really understood.

Thank you for the observation. We removed the tests for more than two groups.

The average duration of the rehabilitation was 101 days. Please add standard deviation.

We added it.

Was the patient, who died, omitted? If not, how can he be included e.g. in the delta Barthel? Are the data presented on 43 or 44 patients?

The data is presented for 44 patients; as Barthel Index is repeated weekly in our unit, the patient who died was also included using the last Barthel score before death.

It makes no sense to include a patient with hypoxic brain damage into the study (spelling error page 4: hypossic), as this patient has no potential for rehabilitation.

We made the spelling correction together with a complete language revision. Really, the patient with hypoxic brain damage made a good (positive) improvement (from Barthel 0 at admission to Barthel 100 at discharge).

The presentation of the data is insufficient. First (table 1) traumatic and non-traumatic patients are regarded separately. Then infections, nutrition and tracheostomy are analysed regardless of the underlying pathology. Trauma patients and vascular patients have to be regarded completely different as they do not share the same prognostic baseline. Trauma patients may have additional injuries that can negatively influence recovery (e.g. head trauma with additional spinal cord injury).

The missing data in the tables are often described in the text; we tried to make tables as simple as possible in order to avoid confusing the reader. We know that many aspects are analyzed regardless of the underlying pathology, but we tried to analyze each factor independently. None of the subjects had additional spinal cord injury (see inclusion and exclusion criteria).

No data about the initial severity of the brain damage is given. Please add GCS after the initial insult to the patients’ characteristics. In addition to this information table 1 should also include at least the number of days in the ICU, the Barthel on admission and discharge from the rehab-centre, the number of patients with tracheostomy on admission and the number of patients with oral feeding.

The initial severity of the brain damage was similar among the patients, as all of them had a Glasgow Coma Scale after the initial insult between 3 and 6. Similarly, the Barthel score on admission to the neurorehabilitation unit was 0 or 5 for all subjects (we added this information in the manuscript). As mentioned before, the missing data in the tables are often described in the text because we tried to make tables more readable.

Figure 2: The average day on the ICU with 62 days seems extremely long. This is unusual and seems unrelated to the disease. What is the real reason? There seems to be a lack of rehabilitation centres, which could take the patients at an earlier point of time. Needs to be discussed. Please give the median of ICU treatment and the range. The correlation coefficient of -0.3 dose not really show an association (should be >-0.5, the p-value does not say anything about the correlation, it just tells us that the calculated value is significant) and might be influenced by the two patients with more than 150 and 200 days of ICU treatment. What happens, when these statistical outliers are eliminated?

The average duration of the hospitalization in the acute care units is surely longer than that reported in the literature. The delay in accessing the Neurorehabilitation Unit of the Conselve Hospital is due to the fact that it is located several kilometers from the Central Hospital. For this reason, many patients with persistent autonomic instability stay in the Hub Hospital for a prolonged period with a delay in access to the rehabilitation unit compared to other centers located in the Hub Hospitals. We added a paragraph in the discussion section. Median time of 57 days was added. The correlation coefficient is not significantly different if the statistical outliers are eliminated.

Table 2: What type of infections? Respiratory (which the reader assumes with the tracheostomy), urinary, blood? All type of infections? Please specify. How many of these patients were still dependent on respiratory ventilation?

 This Table refers to all kinds of infection: the rate of respiratory infections is explained in the text. No patient required respiratory ventilation.

The number of infections is different in table 2 and 4. The data set is always 44 patients and it is assumed, that the same data set for infections is used. In table two the maximum number of infections in any patient is six. In table 4 the maximum number of infections in any patient is three. Explanation?

We apologize for the error; a maximum of 6 infections in now reported also in table 4.

The calculation of the mean number of infections in table 2 cannot be followed. How was it calculated? The reader assumes that the numbers of infections per patient (0, 1, 2, 3 etc.) were summed up, and that this whole number was then divided by the number of patients (n=18). Why does multiplication of 0.2x18 not result in a whole number but 3.6? The same problem can be found in table 4 (e.g., 0,11x11=1,21, which is not a whole number).

The whole number of infections per group was divided only for the subjects who had infections (we added such number of subjects in parentheses in the tables), excluding those who had no infections.

Table 3: What type of infections were observed besides respiratory infections?

The other infections were urinary, blood or gastrointestinal ones (or combined infections). We added this sentence in the manuscript.

There is a lot of confounding parameters which is statistically not analysed. The data show a better effect of treatment in patients without tracheostomy. Is this really surprising? Patients with tracheostomy have more infections. To be expected and well known. That these patients will stay longer in the hospital can also be expected.

For various reasons, our analysis confirms what already known (e.g. better outcomes in patients without tracheostomy). However, the original aspect is represented by the analysis of the rehabilitative intervention in terms of efficiency: this is observed, for example, in the average length of hospitalization, that represents a crucial factor for a better management of the economic and social-health resources.

Discussion: The most common cause for traumatic brain injury is fall.

Thank you for the observation. We added it.

The conclusions presented is common knowledge and, as presented, not supported by the data. A correlation of efficiency of the rehabilitation is not presented (as indicated in the abstract). The data show, that the patients have a different daily improvement when different aetiologies, patients with or without infections and patients with different forms of nutrition are compared.

As already mentioned, we know that, for various reasons, our analysis confirms what is already known. As described in the methods section, we think that the ratio between the functional gain (expressed as the variation of the score on the Barthel Index between discharge and admission) and the duration of the rehabilitative hospitalization can be used as an index of efficiency of the rehabilitation intervention. Such score was compared to all the anaysed variables. Our data suggests, for example, that if a patient has a tracheal cannula for a prolonged period, his/her functional recovery is limited and he/she stays in the hospital for a prolonged period. Consequently, an accurate management of the tracheostomy removal can determine not only a better functional improvement but also a better management of the health resources by guaranteeing shorter hospitalizations.

Reviewer 2 Report

Major issues

It was hard for me to read through this paper in terms of English. One recommendation from me is using a professional English-proof-service.

Even though the abstract was not structured. The results part as a data should be shown.

In the Introduction section, there were many irrelevant topics. The purpose of this study was clearly shown. However, due to the irrelevant and redundant topics, the purpose was made unclear. Even the hypothesis of this study was not shown.

Show what is known and what is unknown in the Introduction section, please. And then, why this study is important should be explained. Without the explanation, nothing is new in this study.

The phrase “ This information can be useful…” is not appropriate in the Introduction part. Too early to say the “useful information” at the beginning of your study. You must show the methodology first.

At the top of the Method section, state ethical approval including study number should be indicated.

To explain Barthel Index, whether it is “excellent” or not is nothing related to this study. Instead of this, explain more and add details of the score for readers who do not know the score.

Again, the methodology is extremely hard to understand. It seems that the primary end point was to compare the Barthel index before and after the intervention of rehabilitation and periods of hospitalization, I think. Secondary outcomes were other factors such as tracheostomy, infection rate, etc. Am I right? And then, furthermore another secondary outcome measurement “ etiology of sABI”. It is tremendously confusing.

In the Statistical analysis section, mean, median, SD, etc. are generally not explained. You mentioned “groups” in this section. However, the groups were not explained in the Method section. Willcoxon test, the par test, Kruskall-Wallis test, Bonferroni method,  Spearman correlation cannot convince readers. Are you really sure this statistical method? Which statistical software did you use? Why was Spearman’s correction needed etc? Since you need more detailed, mathematical explanation here.  

In the Results section, now I understand the groups were “traumatic” vs. “non-traumatic” and “tracheostomy+” vs. “tracheostomy-“. However, the title is “Factors influencing functional recovery during rehabilitation after severe acquired brain injuries”.

Reading through until this part, I understand that you wanted to access the factors influencing functional recovery.

However, to access the factors, you need uni- and multivariate logistic analysis. But the number of the patients with sABI were 14. I understand the number of the patients were too small to apply the appropriate statistical methods.

Since this paper tried to find recovery factors in sABI from 14 patients, I am sorry I do not think you can bring scientifically reliable conclusion even though this was a retrospective study.  

My suggestion is 1. Collecting more patients or 2. Changing methodology completely, for example, tracheostomy plus and minus was interesting point in this study. Hospitalization period was also interesting.

If you have so many patients who were on neuro-rehabilitation, factors that may prolong the period of hospitalization, infection rate, Barthel index etc. might be interesting.

Author Response

We thank the Reviewer for the observations. Below, we explain point-by-point the details of the revisions and the responses to the comments.

It was hard for me to read through this paper in terms of English. One recommendation from me is using a professional English-proof-service.

We made an extensive language revision by a native speaker expert in the medical field.

Even though the abstract was not structured. The results part as a data should be shown.

Due to the limited word count and the great amount of data, we have to limit the description of the results section; anyway, we have added the p values for the correlations we found.

In the Introduction section, there were many irrelevant topics. The purpose of this study was clearly shown. However, due to the irrelevant and redundant topics, the purpose was made unclear. Even the hypothesis of this study was not shown.

Show what is known and what is unknown in the Introduction section, please. And then, why this study is important should be explained. Without the explanation, nothing is new in this study.

The phrase “This information can be useful…” is not appropriate in the Introduction part. Too early to say the “useful information” at the beginning of your study. You must show the methodology first.

In the first part of the introduction section we tried to introduce all the aspects that we analyzed in the following sections and their importance in the rehabilitation pathway. We modified the final part of the section, trying to better explain our hypothesis and the original nature of our study.

At the top of the Method section, state ethical approval including study number should be indicated.

The ethical approval was not required, as our data are collected retrospectively on patients who followed a standard rehabilitation hospitalization and no sensitive data reported in the study could be related to a specific identified subject.

To explain Barthel Index, whether it is “excellent” or not is nothing related to this study. Instead of this, explain more and add details of the score for readers who do not know the score.

We added details of the score; if the reader needs further information, the bibliographic reference can help them.

Again, the methodology is extremely hard to understand. It seems that the primary end point was to compare the Barthel index before and after the intervention of rehabilitation and periods of hospitalization, I think. Secondary outcomes were other factors such as tracheostomy, infection rate, etc. Am I right? And then, furthermore another secondary outcome measurement “ etiology of sABI”. It is tremendously confusing.

The primary endpoint of the study was to identify possible correlations between supposed conditioning factors (etiology of sABI, the time elapsed between the acute event and the rehabilitation hospitalization, the presence of a tracheal cannula, the feeding modalities and the number of infectious episodes) and the index of efficiency of the rehabilitation intervention. Furthermore,  possible correlations between factors were analyzed as secondary endpoints. We inserted in the text primary and secondary outcomes in the Introduction section; additionally, we tried to modify the order of the sentences in methods section in order to make the methodology easier to understand.

In the Statistical analysis section, mean, median, SD, etc. are generally not explained. You mentioned “groups” in this section. However, the groups were not explained in the Method section. Wilcoxon test, the par test, Kruskall-Wallis test, Bonferroni method, Spearman correlation cannot convince readers. Are you really sure this statistical method? Which statistical software did you use? Why was Spearman’s correction needed etc? Since you need more detailed, mathematical explanation here.

The statistical analysis was entrusted to an expert in field that used SAS 9.4 program (SAS Institute Inc., Cary, NC, USA) for Windows. We revised the section removing the test for more than two groups as they were not necessary.

In the Results section, now I understand the groups were “traumatic” vs. “non-traumatic” and “tracheostomy+” vs. “tracheostomy-“. However, the title is “Factors influencing functional recovery during rehabilitation after severe acquired brain injuries”.

Reading through until this part, I understand that you wanted to access the factors influencing functional recovery.

However, to access the factors, you need uni- and multivariate logistic analysis. But the number of the patients with sABI were 14. I understand the number of the patients were too small to apply the appropriate statistical methods.

Since this paper tried to find recovery factors in sABI from 14 patients, I am sorry I do not think you can bring scientifically reliable conclusion even though this was a retrospective study.  

My suggestion is 1. Collecting more patients or 2. Changing methodology completely, for example, tracheostomy plus and minus was interesting point in this study. Hospitalization period was also interesting.

If you have so many patients who were on neuro-rehabilitation, factors that may prolong the period of hospitalization, infection rate, Barthel index etc. might be interesting.

Sorry, we partially disagree. Since the number of subjects we analyzed was 44, the scientific reliablility is different. Anyway, we agree that collecting more patients allows to bring more reliable conclusions.

Round 2

Reviewer 1 Report

Thank you for your anwers to the review, the explanations and the changes in the manuscript. There remain no further questions or comments.

Author Response

Thank you.

Reviewer 2 Report

Much better that the first manuscript. However, still the abstract should be revised. You may use a professional service if you have difficulty with the section. Generally, the abstract consists of Background, Methods, Results and Conclusion.   I guess there are many irrelevant phrases. Therefore, you reached to the words limits. 

Author Response

Thank you for the comments. We revised and modified the Abstract as requested. However, the general structure of this section was not modified since the Istruction for Authors of Trauma Care journal state that “The abstract should be a total of about 200 words maximum. The abstract should be a single paragraph and should follow the style of structured abstracts, but without headings”.